# Stable Superhydrophobic Aluminum Surfaces Based on Laser-Fabricated Hierarchical Textures

**DOI:** 10.3390/ma14010184

**Published:** 2021-01-02

**Authors:** Stephan Milles, Johannes Dahms, Marcos Soldera, Andrés F. Lasagni

**Affiliations:** 1Institut für Fertigungstechnik, Technische Universität Dresden, George-Bähr-Str. 3c, 01069 Dresden, Germany; dahmsjohannes@gmail.com (J.D.); marcos.soldera@mailbox.tu-dresden.de (M.S.); andres_fabian.lasagni@tu-dresden.de (A.F.L.); 2PROBIEN-CONICET, Dto. de Electrotecnia, Universidad Nacional del Comahue, Neuquén 8300, Argentina; 3Fraunhofer-Institut für Werkstoff- und Strahltechnik (IWS), Winterbergstr. 28, 01277 Dresden, Germany

**Keywords:** single-and multi-scale textures, direct laser interference patterning, direct laser writing, superhydrophobicity, aluminum 1050

## Abstract

Laser-microtextured surfaces have gained an increasing interest due to their enormous spectrum of applications and industrial scalability. Direct laser interference patterning (DLIP) and the well-established direct laser writing (DLW) methods are suitable as a powerful combination for the fabrication of single (DLW or DLIP) and multi-scale (DLW+DLIP) textures. In this work, four-beam DLIP and DLW were used independently and combined to produce functional textures on aluminum. The influence of the laser processing parameters, such as the applied laser fluence and the number of pulses, on the resulting topography was analyzed by confocal microscopy and scanning electron microscopy. The static long-term and dynamic wettability characteristics of the laser-textured surfaces were determined through water contact angle and hysteresis measurements, revealing superhydrophobic properties with static contact angles up to 163° and hysteresis as low as 9°. The classical Cassie–Baxter and Wenzel models were applied, permitting a deeper understanding of the observed wetting behaviors. Finally, mechanical stability tests revealed that the DLW elements in the multi-scale structure protects the smaller DLIP features under tribological conditions.

## 1. Introduction

In recent years, environmental protection, resource conservation and energy efficiency have increasingly moved into the social focus [1,2]. As a result, end-user products and manufacturing processes are constantly being optimized to be more resource, energy, and cost-efficient. One way to achieve these targets is designing technical surfaces with enhanced functionalities. Particularly, surfaces with superhydrophobic properties, characterized by a static water contact angle greater than 150° and a contact angle hysteresis below 20° [3,4], have attracted the attention of the scientific community and industrial sector due to their closely related ice-repellent, self-cleaning, and corrosion-inhibiting properties [5].

Superhydrophobic surfaces can be fabricated by coating the material with a low free energy layer, by modifying the surface topography with features in the nano/microscale or by a combination of both approaches [5,6,7,8,9]. Up to now, such superhydrophobic surfaces, also called water-repellent surfaces, have been preferably fabricated by environmentally harmful chemical etching followed by passivation processes, by time-consuming milling methods, by cost-intensive CVD coating methods or by simple multi-steps methods—such as spin, spray, or dip coating—which have a modest throughput limiting their use for large area applications [10,11,12,13,14,15]. Although well-established photolithographic methods have also been employed to generate repetitive microfeatures able to tune the wettability characteristics on different materials, these processes involve multiple cleaning, coating and developing steps, as well as the use of hazardous chemicals and masks [12,16,17].

Laser-based methods represent an innovative solution to produce micrometer sized features on a wide range of materials with a single processing step at high throughputs [18]. Although the direct laser writing (DLW) technique can be considered a state-of-the-art technology, it is however limited in resolution and compromises have to be found to balance resolution and throughput [19,20,21]. On the contrary, the direct laser interference patterning (DLIP) method is able to fabricate micrometer- and even sub-micrometer-sized textures [22,23] at throughputs approaching 1 m^2^/min [24]. The resulting periodic microtextures have shown a promising potential for many different applications, such as improved biocompatibility, increased efficiency in solar cells, or reduced friction and wear in mechanical components [25,26,27,28,29,30].

Due to the broad use of aluminum in the aerospace industry, in the construction sector and in the food processing industry, among other industrial fields, surfaces with enhanced functionalities are on high demand [31,32]. For example, combining nanosecond pulsed laser radiation with a hydrophobic coating, a superhydrophobic microtextured aluminum surface has already been produced [33]. Alternatively, a superhydrophobic nanotextured aluminum surface was fabricated in a multi-step process consisting of etching and coating [34]. Moreover, a superhydrophobic surface was obtained by sandblasting and a subsequent chemical modification on aluminum [35,36]. Artificial anisotropic wetting on aluminum was also explored in the past by producing well-defined grooves by photolithography, that produces physical discontinuities (e.g., sharp solid edge) and chemical heterogeneity on the solid surface [37]. In this direction, also hierarchical micro-/nano-line-like patterns can lead to a preferred orientation of the sitting droplet which further enables the possibility to design surfaces with anisotropic wettability.

It was already demonstrated that single-scale DLIP textured aluminum can provide superhydrophobic and self-cleaning properties [38,39]. Furthermore, multi-scale structures, i.e., textures in which at least two significantly different sized features are combined, have demonstrated an increased delay of ice-formation due to their water-repellent function [40]. However, it is still not clear which textures are more efficient to obtain a superhydrophobic characteristic on aluminum and what role the combination of several textures with different orders of magnitude plays in this context. Moreover, compared to other metals such as stainless steel or copper, aluminum has a comparatively low hardness [41], and therefore, a strategy must be devised to protect the small structures from mechanical damage.

In this work, single and multi-scale structures are produced by nanosecond DLW and picosecond DLIP. The textures are investigated with respect to their topography and morphology. Then, the static long-term as well as dynamic wetting are studied to understand the general wetting behavior. This is supplemented by the application of the well-known wetting models according to Wenzel and Cassie-Baxter theories. Subsequently, the influence of surface chemistry on the wettability characteristics of the laser-treated surfaces is addressed. Finally, the mechanical stability of the single- and multi-scale textures is evaluated from abrasion tests.

## 2. Materials and Methods

### 2.1. Materials

Plates of pure aluminum (EN-AW-1050, SG Designbleche GmbH, Erkelenz, Germany) with lateral dimensions of 100 × 100 mm^2^ and 2 mm thickness were used in this study. The substrates were electrolytically polished resulting in a surface roughness S_q_ of 115 nm ± 10 nm. Before the laser treatment, the samples were cleaned from contamination at 20 °C for 10 min in an ultrasonic bath in isopropanol and afterwards rinsed with distilled water. After the laser processing, the textured samples were stored under atmospheric conditions and no additional treatment was performed.

### 2.2. Laser Surface Texturing

The samples were processed using either nanosecond pulsed DLW or picosecond pulsed DLIP in order to produce the single-scale textures. Multi-scale structures were in turn fabricated by combining both technologies (DLW + DLIP). The DLW process was implemented in a laser surface texturing workstation (GF machining solution P 600, Losone, Switzerland) equipped with a galvanometer scanner system enabling scan speeds up to 1 m/s. The scanner unit includes two mirrors which guide an IR (1064 nm) beam emitted from an ytterbium fiber laser source (YLPN-1000-4x200-30-M, IPG Photonics, Oxford, MS, USA) with a maximal output power of 30 W and an F-theta objective, which focuses the gaussian beam after f = 254 mm onto the substrate (Figure 1a). The pulse duration can be adjusted in the range from 4 to 200 ns. In our experiments, 20 pulses per spot with a pulse duration of 14 ns and at a repetition rate of 30 kHz were applied. The resulting laser fluence was 6.56 J/cm^2^ per pulse. To produce homogeneously distributed DLW textures, the circular laser spots were applied forming a hexagonal array without overlap between adjacent spots (Figure 1b). The distance $S_{DLW}$ between the individual spots was 60 µm and the diameter $d_{DLW}$ of each spot was 50 µm, as shown in Figure 1b.

The used DLIP setup is illustrated schematically in Figure 1c. The laser source was a Nd:YAG solid state laser (neoMOS 70 ps, neoLASE GmbH, Hannover, Germany) with a maximum output power of 2.7 W and a maximum repetition rate of 80 MHz. The pulse duration was 70 ps. The fundamental wavelength of 1064 nm was adjusted to 532 nm in our experiments by a second harmonic generation. Starting from the laser source, a laser beam with a Gaussian power density profile is directed via deflection mirrors onto a diffractive optical element (DOE), which splits the laser beam into four sub-beams of equal intensity. Afterwards, the sub-beams are parallelized by a four-sided prism and then they are superimposed on the substrate surface with a convex focusing lens (focus length: 60 mm) resulting in the intensity distribution expressed by Equation (1) [42]:(1)$$I\left(\vec{r}\right)=\frac{1}{2}\underset{i}{\sum }E_{0i}^{2}+\underset{i<j}{\sum }E_{0i}\cdot E_{0j}\;cos(\left({\vec{k}}_{i}-{\vec{k}}_{j}\right)\cdot \vec{r}+\varepsilon_{i}-\varepsilon_{j}),$$
where $E_{0i}$ is the electrical field, ${\vec{k}}_{i}$ is the wave vector and $\varepsilon_{i}$ is the phase of each of the four sub-beams and $\vec{r}$ is the coordinate vector. Indexes denote the laser sub-beams: 1, 2, 3, and 4 [42]. The distance between the prism and the DOE defines the interference angle *θ* of the sub-beams. The pattern geometry obtained by superimposing these four sub-beams results in a dot-like intensity distribution, whereby the distance between each of individual dot elements in the distribution, or spatial period *Λ*, is determined by the laser wavelength and interference angle *θ* (Figure 1c). The spatial period *Λ* is further calculated by Equation (2) [43]:(2)$$\Lambda =\frac{\lambda }{\sqrt{2}\cdot sin\theta }.$$

By adjusting the incident angle *θ* to 12.78°, 6.35°, and 4.49°, spatial periods *Λ* of 1.7 µm, 3.4 µm, and 4.8 µm were obtained, respectively. The diameter of the interference region $d_{DLIP}$ was ~45 µm for all three spatial periods. In order to structure a homogeneous surface, the individual interference regions were overlapped by 33% in X and Y directions. Thus, in the used processing strategy, the distance $S_{DLIP}$ between the individual interference regions was set to ~30 µm (Figure 1d). The structure depth and morphology were controlled by adjusting the laser fluence in the range 0.36–2.01 J/cm^2^ and the number of pulses per spot were varied between 1 and 15. The pulse repetition rate was set constant at 10 kHz. The laser-texturing experiments were conducted under laboratory conditions with a humidity and temperature of 45% and 21 °C, respectively. No further protective gas was used during the laser processing. To change the focal position, the optical head consisting of the DOE, prism and focusing lens, can be translated in the Z direction. For positioning the substrates in X and Y directions, linear axes (Pro115-05MM-200-UF, Aerotech Inc., Fürth, Germany) were used. These have a maximum travel distance of 200 mm, a maximum speed of 300 m/s, an accuracy of ±8 μm and a resolution of 0.1 μm.

### 2.3. Characterization Methods

The morphology of the fabricated single- and multi-scale micro textures was analyzed using a confocal microscope (Sensorfar S Neox, Terassa, Spain) with a 150× objective resulting in a vertical and lateral resolution of 1 nm and 140 nm, respectively. At least five different areas were measured and the standard deviation was taken as to describe the topography uniformity. The topography was further studied with a scanning electron microscope (ZEISS GeminiSEM 300, Jena, Germany) operating at an acceleration voltage between 3 and 6 kV and a working distance between 8 and 11 mm. For the quantitative analyses of the roughness parameters and the structure height, the software Sensomap 7.3 (Sensofar, Terassa, Spain) was used.

The static water contact angle (WCA) and dynamic wetting measurements (advancing and receding contact angles) were performed using a drop shape analyzer (DSA 100 S, Krüss GmbH, Hamburg, Germany). Deionized water droplets with a volume of 6 µL were positioned on the substrates at a room temperature of 21 °C and 45% relative air humidity. The static water contact angles were calculated using the tangent droplet profile fitting method and each point is the average of at least five measurements. The standard deviation of the measurements was also considered. The dynamic contact angles (advancing and receding WCA) were determined 80 days after the laser treatment using the needle method. Therefore, the droplet volume was inflated and deflated by 2 μL in eight steps of 1 μL/s each per measurement. The value of the advancing and receding WCA of each sample is thus obtained from the mean value of the measured contact angles. The standard deviation of the measurements characterizes the measurement error. The difference of the advancing and receding contact angle determines the WCA hysteresis. The hysteresis is a measure of the adhesion force of a droplet to a surface. With decreasing hysteresis, the adhesion between solid and liquid decreases, whereas the solid–liquid adhesion rises with increasing hysteresis [44,45]. Moreover, the lower the hysteresis of the dynamic contact angle of a surface, the more hydrophobic is the wetting condition [46]. Furthermore, the hysteresis is an important indicator of whether the surface wetting corresponds to the Wenzel- or rather the Cassie-Baxter model. Typically, the contact angle hysteresis is larger in the Wenzel state since the entire surface is wetted and the liquid consequently undergoes an increased adhesion to the structure. In contrast, the reduced solid-liquid contact area in the Cassie-Baxter model allows for less adhesion and therefore minimal hysteresis [44,47].

The mechanical stability of the textures was examined using a ball-on-disc tribometer (Nanotribometer, CSM Instruments, Peseux, Switzerland) operating in linear sliding mode. The normal load was kept constant at 0.4 N and a 6 mm tool steel ball (100Cr6) was used as friction body on the surface. The sliding velocity and the sliding length were set to 5 mm/s and 1 mm, respectively. 300 cycles were used in the experiments and before each measurement the ball and the samples were cleaned with ethyl alcohol (C_2_H_5_OH). The tests were made under ambient conditions with relative humidity of 45% at a room temperature of 23 °C.

## 3. Fabrication of Single- and Multi-Scaled Textures

To address the potential of hierarchical textures on aluminum formed by multi-scaled dot-like patterns to achieve superhydrophobicity with a high chemical and mechanical stability, direct laser writing (DLW) and direct laser interference patterning (DLIP) were combined. First, single-scale textures were produced using both technologies to understand the morphology evolution as a function of the processing parameters and its influence on the wettability characteristics. The targeted DLW texture should not only provide an underlying roughness before the DLIP process to promote a high hydrophobicity, as already demonstrated in earlier publications [38,48], but also it should serve to protect the smaller features produced by DLIP [49].

The DLW produced pattern morphology consists of individual elements with a diameter of 50 µm, which were distributed in a hexagonal array (Figure 2a). The application of 20 pulses per element (with 14 ns of pulse duration) resulted in a stepwise mechanism of melting and resolidification of the material within the irradiated area. Two characteristic processes are decisive for the structure formation. As it is very well known, the nanosecond pulses causes the local melting of the material at the irradiated regions due to the relative large heat affected zone [50,51]. Then, the resulting melting pool in the center of the laser spot produces a smooth surface after solidification (inset Figure 2a). In addition, the molten material flows to the edge of the ablation area by a combination of the recoil pressure (which is also described in the literature as a piston mechanism [52]) as well as Marangoni convection [53]. By applying several laser pulses at the same position, different layers of resolidified material are obtained, resulting in a characteristic honeycomb geometry (see insert in Figure 2a).

After that, four-beam DLIP technology was used to fabricate significantly smaller features. In order to fabricate multi-scale textures, the same specimen which was previously textured with DLW, was afterwards textured with DLIP. A picosecond pulsed laser source was used to minimize the influence of molten material on the final microstructure. This can be explained by the very short thermal diffusion length of 80 nm, which can be calculated for a pulse duration of 70 ps. The objective was to generate dot-like textures at least one order of magnitude smaller than the DLW textures, and thus three different spatial periods, namely 4.8 µm, 3.4 µm, and 1.7 µm were used. Each of these periods was processed with low and high number of pulses in order to obtain shallow and deep textures. The selected laser fluences were taken from previous parameter screening experiments. Table 1 provides an overview of the laser parameters used for DLIP texturing.

An exemplary 3.4 µm shallow texture is shown in Figure 2b. Due to the optimized spacing during the manufacturing process (see Section 2.2) the surface is homogeneously patterned with the dot-like elements. Nevertheless, there are minor deviations in the homogeneity due to the Gaussian energy distribution of the laser beam. This leads to a higher energy density in the center of the interference area than in the peripheral areas and thus, to deeper structures in the center of the laser spot. The inset in Figure 2b shows a symmetrical distribution of four individual DLIP dots.

Next to each dot, unprocessed material is observed (yellow colored area in Figure 2b). There regions correspond to the interference minima and therefore the laser energy is too low to melt or ablate the material. Furthermore, it is nearly unaffected by melt ejections solidifications, which is also a consequence of using ps pulsed laser radiation on aluminum [51]. Figure 2c shows the processed Al surfaces with deep DLIP features and a spatial period of 3.4 µm. In contrast to the shallow DLIP structures the distribution of the microstructures became less homogeneous, compared to the shallow DLIP texture, as a considerable number of craters collapsed forming large and irregular side walls and hills (yellow circles in Figure 2c). Furthermore, the amount of unprocessed material between the individual DLIP dots was strongly reduced.

Hierarchical topographies consisting of multi-scaled elements were produced by combining DLW and DLIP methods. The Al-surface was first treated with the DLW technique and then with DLIP, using the same spatial periods as described above. For the fundamental DLW structure the process parameters presented in the beginning of Section 3 were used. The employed DLIP parameters for the fabrication of the multi-scale textures are taken from the single-scale experiments reported in Table 1.

After DLIP structuring, the DLW craters were completely covered by the DLIP dots. The resulting multi-scale topographies are shown in Figure 3. It can be clearly seen that both the shallow (Figure 3a) and the deep (Figure 3b) 1.7 µm DLIP textures have only a marginal influence on the DLW topography. The molten material, which moves to the edge of the single DLW crater forming a ‘corona’, is still visible. Similar results can be seen for the multi-scale structures with a DLIP period of 3.4 µm (Figure 3c,d). In the shallow DLIP textures, isolated melt droplets appeared in the DLIP dots (arrows in insert Figure 3c). These were much more pronounced in the deeper DLIP 3.4 µm structures and resulted in the formation of a nano-roughness with feature sizes between 100 and 300 nm between the individual DLIP dots. The droplets were pushed out of the center of the dots and piled up on the narrow bridges between the dots (yellow framed area in insert Figure 3d). Thus, a three-scale texture was fabricated from DLW (first scale, 60 µm spatial period), DLIP dots (second scale, spatial period 3.4 µm) and the self-organized nano-roughness (100–300 nm).

Figure 3e,f show the topography of the multi-scale textures featuring DLIP patterns with a period of 4.8 µm, with low and high accumulated fluences, respectively. In both cases, a homogeneous DLIP dot-like texture was observed over the larger pattern. Furthermore, the multi pulse ablation of both shallow and deep DLW+DLIP structures led to complete smoothening of the edges of the DLW spots. The individual layered crater walls are no longer visible (Figure 3e,f). Subsequently, the structure depth is decreased compared to DLW textures (Appendix A). The deeper 4.8 µm DLIP texture exhibit also a nano-roughness, but it is not as pronounced as in the 3.4 µm textures (insert Figure 3f), possibly due to the larger spatial period of 4.8 µm and thus the larger area between the dots. Thus, the droplets are more likely to spread across the narrow bridges between the dots (insert Figure 3f).

The topography of the structured surfaces was also characterized by confocal microscopy. The structure depths for the fabricated textures, single-scale as well as multi-scale, are given in Appendix A along with the used process parameters. Figure 4 summarizes the structure depths of the individual DLIP and DLW features in each hierarchical surface. Due to the location-dependent DLIP depth within the DLW structure, this can only be determined to a limited extent. Therefore, it was arbitrarily defined that all measurements were carried out in the central area of the DLW crater. From Figure 4, it can be observed that the maximum structure depth of the hierarchical textures was slightly decreased by the DLIP processing. Due to the ablation process mentioned above, the DLW structure depths decrease to values ranging from 27.9 µm (DLW+DLIP_4_._8µm deep_) to 29.1 µm (DLW+DLIP_1_._7µm shallow_). The inserts in Figure 4 show confocal microscopy images taken on the hierarchical textures with different DLIP depths.

## 4. Characterization and Modeling of the Surface Wettability

### 4.1. Static Long-Term and Dynamic Wetting Analysis

The evolution of the wetting properties of the laser-textured surfaces was examined by means of static water contact angle (WCA) measurements performed one day after laser structuring and during a period of about 30 days. To estimate the long-term development of the wetting characteristics, the contact angles of the structures were measured again after 80 days. The WCAs of the fabricated structures (colored symbols) and untreated reference sample (black squares) as a function of time are depicted in Figure 5. The contact angle evolution of the single-scale DLIP structures is shown for the shallow homogeneous topographies (Figure 5a) and deep inhomogeneous topographies (Figure 5b). Additionally, the evolution of the WCA of the multi-scale topographies with shallow and deep DLIP structures is shown in Figure 5c,d, respectively. Each diagram also includes the droplet shape of the untreated reference surface and the shape of the droplet on the surface with the highest WCA. The WCAs of the untreated reference surface averaged 95°.

Over the period of the measurements, all fabricated topographies showed an increase in the WCA with time, starting with hydrophilic wetting during the first days, up to stationary hydrophobic and in some cases even superhydrophobic state at the end of the measurement period.

The shallow homogeneous DLIP structures have shown different WCA evolution, depending on the spatial period. With increasing period (and with the corresponding increased structure depth), the laser treated Al surfaces showed a less pronounced WCA increase with time. For example, the hydrophobic condition was achieved with the DLIP_1_._7µm shallow_ after 4 days and with DLIP_4_._8µm shallow_ after 12 days (Figure 5a). The transient behavior observed in all the laser-processed samples can be explained by the presence of an oxide layer (Al_2_O_3_) on the aluminum surface formed during the laser treatment due to thermal effects [54], with a typical thickness between 2 and 20 nm, as reported elsewhere [55,56]. This oxide layer features polar regions of unsaturated oxygen compounds that provide a high surface energy inducing a hydrophilic wetting immediately after the laser treatment and during the first days. Later, organic molecules from the ambient air accumulate on the substrate surface, which leads to an increase of WCA [45,57]. This effect was already observed in earlier experiments and confirmed by chemical surface analyses [57,58]. In those studies, the microstructures were analyzed 2 and 20 days after the laser processing using XPS measurements, revealing an increased carbon content on the surface. Immediately after laser processing, a hydrophilic characteristic was originated by polar sites composed of unsaturated aluminum and oxygen atoms. However, due to long-chain hydrocarbons with carbonyl groups in the atmosphere, the carbon content on the surface increased. This resulted in a decrease of the polarity and contributed consequently to the hydrophobic component of the WCA [54,58,59,60]. Simultaneously, the roughness linked to the high structure depth of the texture with a spatial period of 4.8 µm lead to higher WCAs (e.g., after 30 days: 137° for DLIP_4_._8µm shallow_ and 106° for DLIP_1_._7µm shallow_). It can be assumed from this result, that DLIP textures with an increased structure depth (i.e., DLIP_1_._7µm shallow_: 0.6 μm vs. DLIP_4_._8µm shallow_: 3.2 μm) are able to retain more air in their structural cavities. Then, according to the Cassie-Baxter wetting model, the contact area of solid and water droplets is minimized, resulting in a higher WCA [61]. In comparison, all deep DLIP topographies show a similar behavior, reaching higher WCAs than the shallow structures up to values around 145° (Figure 5b).

The WCA evolution of the shallow and deep multi-scale topographies is shown together with the pure DLW textures in Figure 5c,d. It can be seen that, in general, the multi-scale structures reach the hydrophobic state faster than the DLW textured surfaces. With respect to the long-term evolution (80 days), all multi-scale topographies as well as the DLW textures achieved a superhydrophobic condition (WCA > 150°). The change in depth of the DLIP structures in the hierarchical textures (0.3–1.5 μm, see Figure 4) as well as in the absolute structure depth of the hierarchical topographies (29.1–28.3 μm, see Figure 4), seem to have a minor influence on the wettability, since the final WCA of all hierarchical structures is nearly the same.

The deep multi-scale structures followed a similar temporal evolution (Figure 5d) as the shallow hierarchical structures, except for the DLW+DLIP_3_._4µm deep_ topography. This sample turned hydrophobic (WCA > 90°) after just 4 days and reached on day 10 the superhydrophobic condition with a WCA of ~160°. This wetting state was also confirmed after 80 days. The reason could be related to the nano-roughness described in the topography analysis (Figure 3d) which was observed exclusively on this topography. These random structures with a feature size between 100 and 300 nm exhibit a very low capillary adhesion due to the absence of sealed air pockets [62,63]. Hence, in this area the air is not locally trapped, as assumed in the Cassie-Baxter model. Subsequently, the nano-texture is able to stabilize the liquid contact line between droplet and solid surface [64], preventing a transition to the Wenzel state and thus maintaining the superhydrophobic condition [65,66]. An overall comparison of the single- and multi-scale topographies suggests that the hierarchical structures exhibit a high water repellence (and higher WCA, accordingly) developed after a shorter time.

For a better characterization of the wetting performance, dynamic measurements based on the advancing and receding WCA were also conducted. The difference between these values results in the WCA hysteresis, which represents another fundamental wetting parameter. The causes of hysteresis are primarily surface roughness and a heterogeneous surface chemistry [4,45]. Figure 6 shows the static, advancing, and receding WCA and the hysteresis values of the untreated and treated Al samples. The labels in the *x*-axis represent the different spatial periods of the DLIP microstructures, for the single- or multi-scale textures.

Several clear trends are visible from this figure: with increased static WCA (red bar), which lies in all cases between the advancing (green) and the receding (violet) WCA, the hysteresis decreased (yellow bar). Thus, the untreated reference surface, which has the lowest static WCA of 95°, presents a maximum hysteresis of 26°. Interestingly, a correlation between the spatial period Λ of the DLIP process and the final WCA can be observed in the shallow single-scale DLIP structures (Figure 6a), as these patterned surfaces showed an increasing WCA with increasing spatial period. This effect can be explained by the increased roughness. For the deep single-scale structures, the DLIP spatial period seems to play a minor role, since all textures showed a hydrophobic (90° < WCA < 150°) but not superhydrophobic characteristic. This indicates that the influence of the chemical surface modification due to the increased laser fluence and pulse number is dominant (Figure 6b). Similar results were obtained with the combination of DLW with shallow DLIP patterns.

As all shallow multi-scale surfaces exhibit similar WCA and hysteresis values similar to the DLW texture (Figure 6c), it can be concluded that the topography modification induced by the shallow DLIP structures is too low to have a significant effect on wetting. For these hierarchical textures, the WCA and hysteresis were nearly constant around the values of 150° and 17°, respectively. In contrast, the hierarchical textures featuring deep DLIP structures have shown the largest values of WCA and lowest hysteresis. Particularly, the multi-scale surface structured with a DLIP spatial period of 3.4 µm reached a WCA and a hysteresis of 163° and 8.9°, respectively, thus standing out from all other topographies (Figure 6d). It is assumed that this superhydrophobic state is mainly caused by its unique hierarchical texture, consisting of three levels—i.e., DLW, DLIP, and nano-roughness—between DLIP craters (insert Figure 3d).

### 4.2. Modeling of Wetting Behavior

In the previous sections, the wetting models according to Cassie-Baxter and Wenzel theories were presented. In this section, these two models are used to calculate the WCA of the rough laser-textured surfaces and compared to the experiments corresponding to day 80. The geometrical models used for the calculation of the WCA according to Wenzel and Cassie-Baxter are shown in Appendix A, as well as their corresponding equations to calculate the roughness parameters (Appendix A). Although these two models consider only the geometrical shape of the solid surface and not the modification of the surface chemistry upon surface texturing, the initial surface chemistry of the flat sample are taken into account in these models by the inclusion of Young’s contact angle, which describes the interplay between the surface energies between solid-vapor, solid-liquid, and liquid-vapor interfaces [67,68,69].

Applying the Cassie-Baxter and Wenzel models for the shallow single-scale structures, the calculated WCAs showed an upward trend as the spatial period increased, in agreement with the measured WCAs (Figure 7a). Particularly, the Cassie-Baxter model seems to fit better to the experimental results than the Wenzel model, suggesting the presence of air cushions between the surface and liquid. In contrast, the calculations derived from both models differ strongly from the measured DLIP_deep_ and the DLW textures, which all have a WCA > 140°. As the used models are based on geometrical parameters, the observed high WCAs might be a consequence of the presence of the carbon adsorbed in the surface as discussed by several authors [33,38,54,57,58,63]. Therefore, the limited WCAs predicted by Cassie-Baxter and Wenzel models suggest that for these textured samples the topography plays a minor role to explain the increase in the WCA.

The simulation results of the multi-scale textures present a different situation. Assuming the Wenzel wetting model, these surfaces should provide a low WCA of about 103°, which differs significantly from the measured values. Differently, if Cassie-Baxter wetting is assumed, the calculated WCAs vary between 138° and 151°, agreeing very well with the measured WCAs for the hierarchical textures with all shallow DLIP structures as well as for the DLW + DLIP_1_._7µm deep_ sample (Figure 7b). The surface chemistry plays also a role on these multi-scale textures, although its influence seems to be negligible compared to the contribution of topography on the measured WCA. Interestingly, the difference between the WCAs predicted by the Cassie-Baxter model and the measured values in the DLW + DLIP_3_._4µm deep_ and DLW + DLIP_4_._8µm deep_ textures is significantly large. These differences can be attributed to the fact that the adopted Cassie-Baxter model does not include the nano-scale topography observed in these samples, because of the impossibility to measure the actual heights of the random nano-structures with the used confocal microscopy.

### 4.3. Mechanical Stability of Hierarchical Structures

To prove the ability of the large DLW textures to protect the DLIP structures, abrasion tests were performed on a hierarchical DLW + DLIP_3_._4µm deep_ sample using a ball-on-disk tribometer (see Material and Methods section). For comparison purposes, a DLIP_3_._4µm deep_ treated sample was also evaluated at the same abrasion conditions. The WCAs were 148° and 163° for the DLIP and DLW + DLIP texture, respectively before the abrasion test. The resulting structure depth of the single- and the multi-scale pattern was 3.9 µm and 28.0 µm, respectively. SEM images of these samples after the abrasion tests are depicted in Figure 8a,b, respectively. It can be observed that the soft aluminum deforms under the pressure of the hard steel ball (red colored area in Figure 8a), which crushes the single-scale DLIP texture and only marginal areas of the DLIP texture remain. In contrast to that, the DLIP patterns inside the DLW craters were not damaged over a large area and only the edges of the DLW crater were worn off (red colored area in Figure 8b).

After the abrasion tests, the corresponding WCAs of the DLIP_3_._4µm deep_ and DLW+DLIP_3_._4µm deep_ textures were reduced to the values 143° and 150°, respectively, as shown in the insets of the Figure 8a,b. This suggests that the superhydrophobicity is reduced by the destruction of the texture. However, for the hierarchical structures, it was preserved.

## 5. Summary and Conclusions

Aluminum 1050 substrates were treated using nanosecond pulsed (14 ns) DLW and picosecond pulsed (70 ps) DLIP processes in order to fabricate periodic single-scale and multi-scale surface textures. The DLW structures, used for the larger scale of the hierarchical textures, had a diameter and depth of approximately 60 µm and 30.1 µm, respectively. Using four-beam DLIP, dot-like structures with a spatial period of 1.7 µm, 3.4 µm, and 4.8 µm were produced with different depths ranging from 0.6 µm to 5.3 µm.

The wettability properties of the samples were examined over a period of 80 days. After an initial hydrophilic behavior, all laser-treated substrates showed a transition into the hydrophobic or even superhydrophobic condition. It was found that in addition to the depth of the DLIP textures, also chemical changes on the surface influenced the final wetting state. Particularly, the multi-scale texture DLW + DLIP_3_._4µm deep_ showed a superhydrophobic behavior with a WCA of 163°, which was the largest among all the samples. This characteristic could be related to the presence of a third scale in the form of a nano-roughness with a feature size in the order of tens of nm. Calculations using the Wenzel and Cassie-Baxter models revealed that they are only of limited use for determining the wetting state of single-scale textures. For multi-scale textures, however, a strong correlation of the measured WCA with the WCA calculated according to Cassie-Baxter theory was found.

Additionally, wear tests were performed in order to study the mechanical stability of the textures. The experiments showed that the single-scale DLIP_3_._4µm deep_ texture were completely destroyed, whereas the DLW component in the multi-scale DLW + DLIP_3_._4µm deep_ texture protected the smaller DLIP features inside the craters. Furthermore, on the DLW + DLIP texture, the superhydrophobic property was retained with a WCA of 150°. Therefore, it can be concluded that the hierarchical texture allows producing multifunctional surfaces, e.g., increasing water contact angles as well as improving the mechanical stability of the patterns.

## Figures and Tables

**Figure 1 materials-14-00184-f001:**
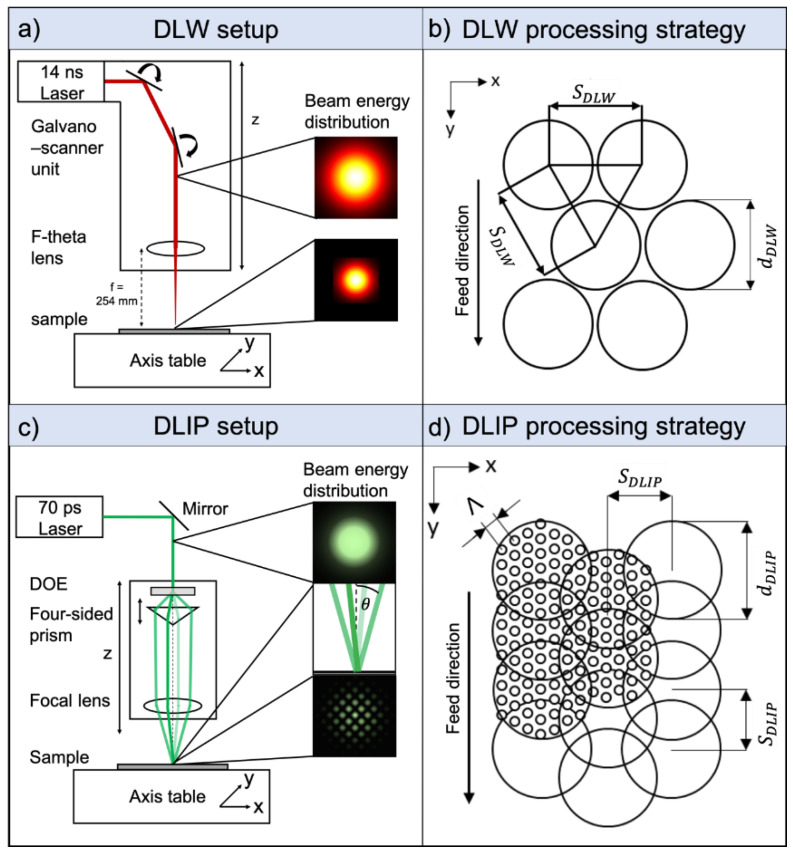
Schematic drawing of the used DLW setup, the local laser beam energy distribution (**a**) and the corresponding processing strategy (**b**). The used DLIP setup including the optical elements and the four-beam interference energy distribution (**c**) and the DLIP processing strategy depicting the dot-like interference pattern in the focal zone (**d**) are shown.

**Figure 2 materials-14-00184-f002:**
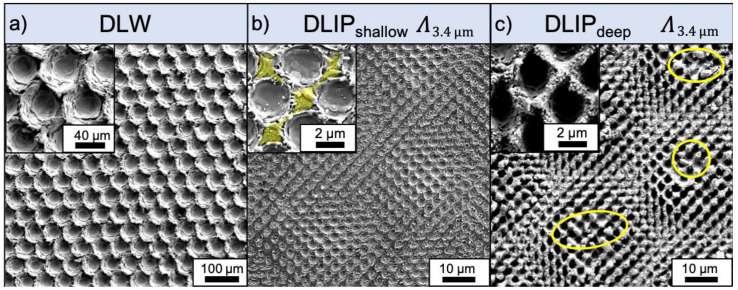
SEM images of the single-scale DLW and DLIP textures with a spatial period of 60 µm and 3.4 µm, respectively. The hexagonally arranged dot-like DLW textures were fabricated using a laser fluence of 6.56 J/cm^2^, 20 pulses per dot and a repetition rate of 30 kHz. The pulse duration of the laser was 14 ns and the used wavelength was 1064 nm (**a**). The DLIP_shallow_ texture was fabricated using 2 pulses per DLIP area and a laser fluence of 0.82 J/cm^2^ (**b**), the DLIP_deep_ texture was fabricated using 15 pulses per DLIP area and a laser fluence of 1.33 J/cm^2^ (**c**). The laser wavelength, the pulse duration and the repetition rate were kept constant at 532 nm, 70 ps, and 30 kHz, respectively.

**Figure 3 materials-14-00184-f003:**
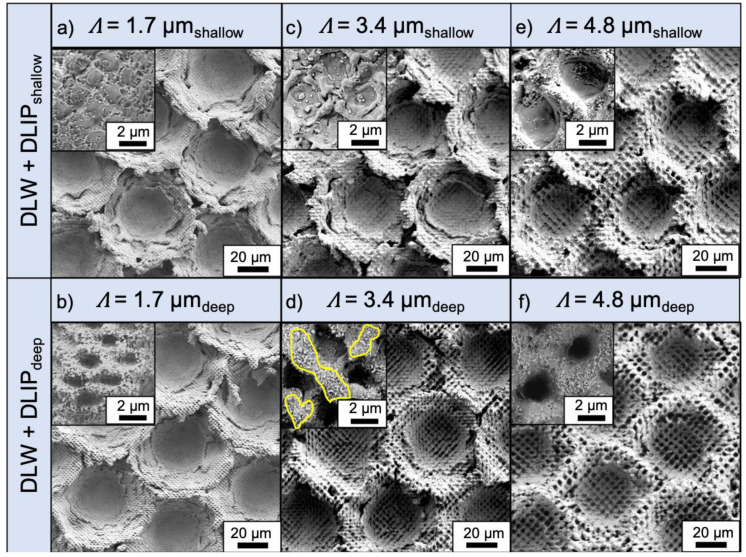
SEM images of the multi-scale textures resulting from DLW combined with a shallow DLIP pattern with a spatial period of 1.7 µm (**a**), 3.4 µm (**b**), and 4.8 µm (**c**) as well as the combination of DLW and deep DLIP pattern with the same spatial periods (**d**–**f**), respectively. For the DLW processing 20 pulses were used, each with a laser fluence of 6.56 J/cm^2^. Details about the number of pulses and the laser fluence for the DLIP processing are given in Table 1

**Figure 4 materials-14-00184-f004:**
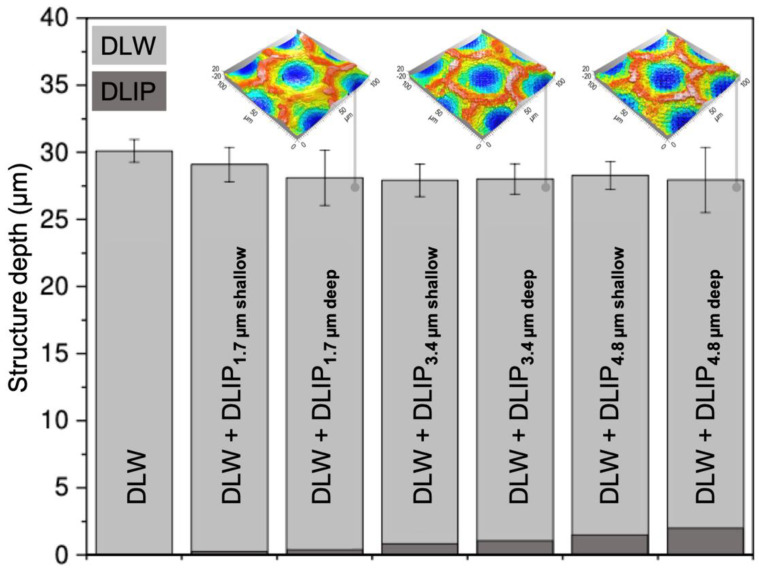
Overview of the structure depths of the DLW structure, the multi-scale DLW+DLIP structures, the DLIP structures within the DLW+DLIP textures as well as exemplary topography images taken with confocal microscopy of the deep multi-scale textures.

**Figure 5 materials-14-00184-f005:**
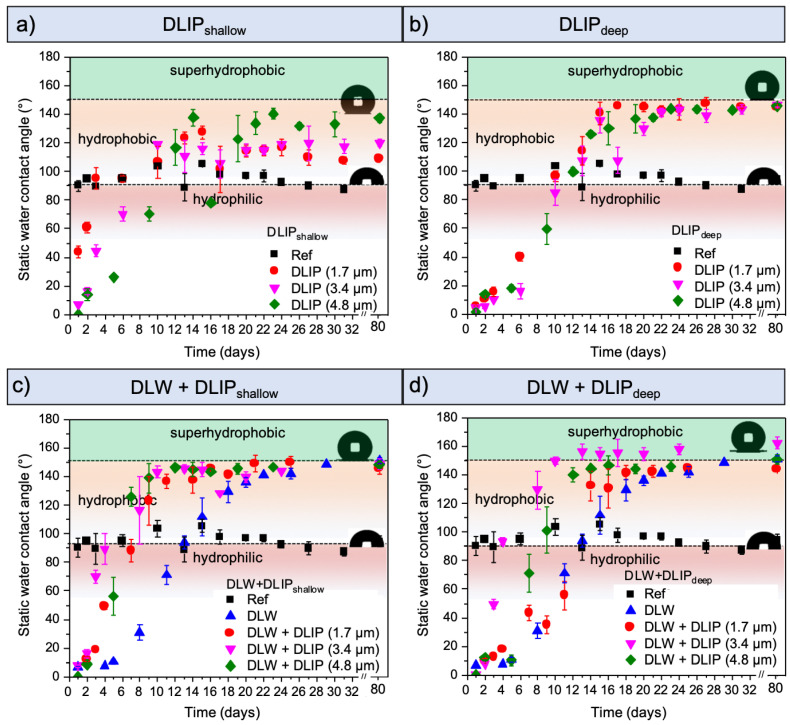
Evolution of static water contact angles for shallow homogeneous (**a**) and deep inhomogeneous (**b**) DLIP structures as well as multi-scale and DLW-containing structures with shallow (**c**) and deep (**d**) DLIP structure depths. The DLIP spatial period, in µm, is indicated by the number in the square brackets.

**Figure 6 materials-14-00184-f006:**
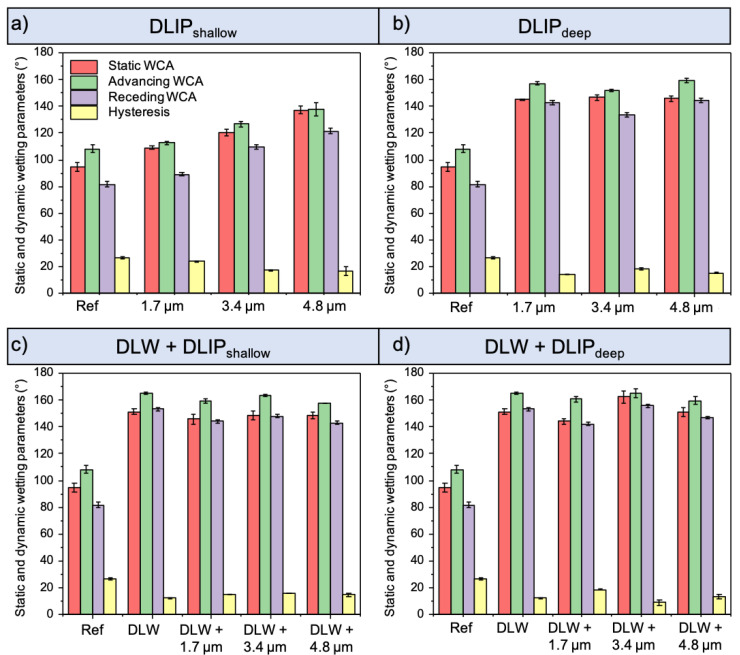
Static, advancing and receding water contact angles as well as the WCA hysteresis of the fabricated single-scale shallow (**a**) and deep (**b**) DLIP textures and of the multi-scale textures including shallow (**c**) and deep (**d**) DLIP elements measured 80 days after laser processing.

**Figure 7 materials-14-00184-f007:**
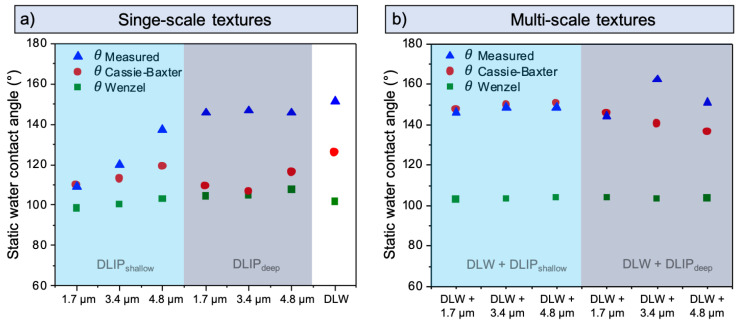
Static WCA calculated according to the Cassie-Baxter and Wenzel models and measured after 80 days for the produced single-scale structures (**a**) and for the multi-scale structures (**b**). The labels in the horizontal axes represent the spatial period of the DLIP textures.

**Figure 8 materials-14-00184-f008:**
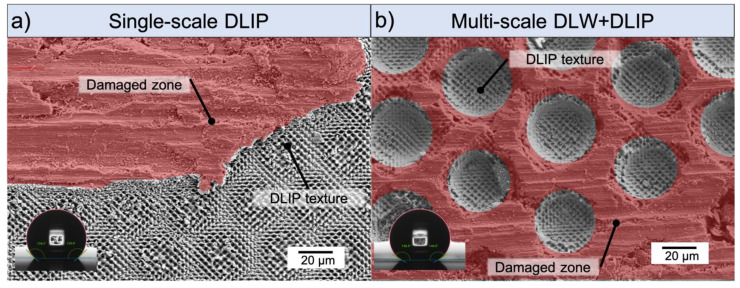
SEM images of wear tracks achieved by a ball-on-disk test on a single-scale DLIP_3_._4µm deep_ texture (**a**) and on a multi-scale DLW + DLIP_3_._4µm deep_ treated texture (**b**). The red colored areas indicate the damaged zone. The used 100Cr6 ball had a diameter of 6 mm and the normal load was kept constant at 0.5 N.

**Table 1 materials-14-00184-t001:** Applied number of pulses and laser fluence for shallow and deep dot-like DLIP textures

	DLIP Λ=1.7 µm		DLIP Λ=3.4 µm		DLIP Λ=4.8 µm	
Parameter (Unit)	Pulses (#)	Fluence (J/cm^2^)	Pulses (#)	Fluence (J/cm^2^)	Pulses (#)	Fluence (J/cm^2^)
DLIP_shallow_	1	0.36	2	0.82	3	2.01
DLIP_deep_	10	0.56	15	1.33	15	1.58

## Data Availability

Not applicable.

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
