# Peer review of "Stable Superhydrophobic Aluminum Surfaces Based on Laser-Fabricated Hierarchical Textures"

_materials, 2021, doi:10.3390/ma14010184_

Round 1
Reviewer 1 Report
This is an interesting paper aimed to assess the effect of single and multi-scale structures on the superhydrophobic behaviour of aluminium surfaces. The paper is well structured and the experimental campaign is clearly explained.
Some remarks, in my opinion require further discussion.
-
The Authors address the growing superhydrophobic behaviour of the samples to the interaction of airborne molecule with aluminium substrate during time. This aspect, should be corroborated by experimental evidence. Furthermore, should be useful to better clarify the coupled role of surface energy modification and micro/nano roughness.
-
On the 4.2 section a modelling of the wetting behaviour was proposed. Considering the regular surface morphology this approach is plausible. Although one point of discussion, that need to better explained, is related to the effective fitting results observed by Cassie-Baxter model for single and multi-scale texture. Despite multi-scale texture, on single scale texture, the fitting model is not effective. The Authors need to better justify these results. The results , as indicated, are influenced by surface texture (rows 396-397). However a contribute is related to the presence of the carbon adsorbed in the surface (rows 381-382). Seems that this latter contribute is relevant in Figure7a batches and less relevant for Figure 7b one. A clarification of the role of these contributes on these structures need to be added.
-
Based on rows 410-413, the surface abrasion not influences significantly the WCA (roughly a similar percentage of reduction should be calculated). This should suggest that the carbon adsorbed in the surface (abraded during the test) not have a relevant role on the superhydrophobic behaviour of the samples. At the same time, the roughness is also modified with a consequent effect on the properties of its surface. This is a critical point, which, although it is not the main topic of this work, represents an aspect that must at least be addressed already on this context.
Minor remarks:
- The introduction could be extended adding further information on the role of micro- and nano- roughened geometries on superhydrophobic behaviour of aluminium alloys.
- Figure 7a. Correct with “Single-scale”
Reviewer 2 Report
This is a thorough and profound study on the preparation and characteristics of structured aluminium surfaces with a view to superhydrophobic surfaces. In fact, the structures are well defined and well documented. The manuscript is fluently and clearly written and well illustrated with a number of images of very high quality. The results are clear and the discussion is sound. I do not have significant problems with this manuscript and wold like to address only three minor remarks:
Line 41 – 44: The authors write: “Up to now, such superhydrophobic surfaces, also called water-repellent 42 surfaces, have been preferably fabricated by environmentally harmful chemical etching followed by 43 passivation processes, by time-consuming milling methods or by cost-intensive CVD coating 44 methods”. However, it should be noted that there are also simple methods to prepare e.g. superhydrophobic surfaces based on cellulose, as described e.g. in the following review: Hannu Teisala, Mikko Tuominen, Jurkka Kuusipalo, Superhydrophobic Coatings on Cellulose‐Based Materials: Fabrication, Properties, and Applications, Adv. Mater. Interfaces 2014, 1, 1300026. Moreover, many synthetic polymers with superhydrophobic surfaces can readily be prepared, see e.g Iskender Yilgor, Sevilay Bilgin, Mehmet Isik, Emel Yilgor, Facile preparation of superhydrophobic polymer surfaces, Polymer 2012, 53, 1180.
Line 79: It is unclear to which confidence level the deviation of 10 nm refers to. Does this value reflect a 95% confidence level or is this the difference between the extreme values?
Line 375 – 397: The differences in the fitting of the experimental results with the models is not worrying at all and can simply be due to the limitations of the models which are based on a number of prerequisites which may not be met perfectly in reality.
Reviewer 3 Report
Stable superhydrophobic aluminum surfaces based on laser-fabricated hierarchical textures
2.2 laser surface texturing
In description of DLW, DLIP process you could add in which atmosphere it was performed: air or protective gas?
- fabrication of single and multi-scaled textures
Line 186: after that….it is not clear! You meant, you took another specimen to perform DLIP…not on the same specimen.
Table 1 and Table 2 are the same. You could show parameters in one table similar as in Table S1.
4.1 static long-term and dynamic wetting analysis
Line 295: have you measured the thickness of Al2O3 layer? Is at the surface of the reference specimen also Al2O3?
Line 341: CA or WCA? What is CA?
4.2 modelling of wetting behavior
Line 382: from where did carbon adsorb in the surface?
4.3. mechanical stability of hierarchical structures
Line 411: do you mean 143 mm or 143°?
Line 413: what was structural depth change after abrasion tests? Could you express in % the change of structure under the droplet (before and after abrasion test: before abrasion you have 100% of hierarchical topography, what is % of hierarchical topography after abrasion test?)
